# An Exploratory Study on the Physical Activity Health Paradox—Musculoskeletal Pain and Cardiovascular Load during Work and Leisure in Construction and Healthcare Workers

**DOI:** 10.3390/ijerph19052751

**Published:** 2022-02-26

**Authors:** Suzanne Lerato Merkus, Pieter Coenen, Mikael Forsman, Stein Knardahl, Kaj Bo Veiersted, Svend Erik Mathiassen

**Affiliations:** 1Research Group for Work Psychology and Physiology, National Institute of Occupational Health, Pb 5330 Majorstuen, 0304 Oslo, Norway; bo.veiersted@stami.no; 2Department of Public and Occupational Health, Amsterdam Public Health Research Institute, Amsterdam UMC, Vrije Universiteit Amsterdam, 1081BT Amsterdam, The Netherlands; p.coenen@amsterdamumc.nl; 3Division of Ergonomics, School of Engineering Sciences in Chemistry, Biotechnology and Health, Halsovägen 11C, 141 57 Huddinge, Sweden; miforsm@kth.se; 4Institute of Environmental Medicine, Karolinska Institutet, Box 210, 171 77 Stockholm, Sweden; 5Research Strategy, National Institute of Occupational Health, Pb 5330 Majorstuen, 0304 Oslo, Norway; stein.knardahl@stami.no; 6Centre for Musculoskeletal Research, Department of Occupational Health Sciences and Psychology, University of Gävle, 801 76 Gavle, Sweden; svenderik.mathiassen@hig.se

**Keywords:** cardiovascular load, physically demanding work, musculoskeletal disorders, compositional data analysis, healthcare sector, construction industry, occupational physical activity, leisure time physical activity

## Abstract

Using a novel approach, this exploratory study investigated whether the physical activity (PA) paradox extends to cardiovascular load and musculoskeletal pain. At baseline, 1–2 days of 24 h heart rate was assessed in 72 workers from construction and healthcare. Workers then reported pain intensity in 9 body regions (scale 0–3) every 6 months for two years. The 2 year average of musculoskeletal pain (sum of 9 pain scores; scale 0–27) was regressed on time spent during work and leisure above three thresholds of percentage heart rate reserve (%HRR), i.e., ≥20 %HRR, ≥30 %HRR, and ≥40 %HRR, using a novel ilr structure in compositional data analysis. Analyses were stratified for several important variables. Workers spending more time in physical activity at work had higher pain, while workers with more time in physical activity during leisure had less pain (i.e., the PA paradox), but none of the associations were statistically significant. Higher aerobic capacity and lower body mass index lowered the pain score among those with higher physical activity at work. This exploratory study suggests that the PA paradox may apply to musculoskeletal pain and future studies with larger sample sizes and additional exposure analyses are needed to explain why this occurs.

## 1. Introduction

A growing body of studies have reported that the generally accepted positive health effects of physical activity do not hold for both leisure and work. While it has consistently been shown that regularly engaging in high-intensity leisure-time physical activity (LTPA) is associated with positive health effects, occupational physical activity (OPA) with high intensity is associated with no beneficial, or even negative, health effects. This phenomenon, which is referred to as the ‘physical activity paradox’ [1], has been shown for various health outcomes such as cardiovascular health [2,3,4], some cancers [3,5], sleep quality [3], and all-cause mortality [6].

Although studied to a lesser extent, there are indications that the physical activity paradox also extends to musculoskeletal disorders. Here, LTPA may contribute to reducing the risk of musculoskeletal disorders [7,8,9,10,11], while OPA has been reported to increase the risk of musculoskeletal disorders in the low back, neck/shoulder and lower extremities [12,13,14]. 

In the current debate regarding the physical activity paradox, it is still an open question why the health effects would differ between OPA and LTPA. One explanation could be that people have different modes of physical activity at work and during leisure. LTPA often occurs during relatively short activities such as walking, running, swimming, or cycling. In contrast, for OPA, factors such as manual handling, lifting, and prolonged awkward postures are often considered [15]. Such OPA exposures are typically of a long duration, and at an intensity below the physiological threshold for improvement or maintenance of health [16]. OPA and LTPA may, therefore, differ substantially regarding duration, frequency, intensity, and, thus, recovery opportunities, which would lead to differential outcomes [17]. Alternatively, there could, indeed, be differential health effects of the same exposure, depending on whether it is performed at work or during leisure. A reason for this could, for example, be that additional factors, such as psychosocial work demands, could elevate heart rate [18] and influence health outcomes [19,20]. An appropriate design for investigating potential causes of the paradox would be to investigate the impact of various intensity levels of OPA and LTPA, and to compare OPA and LTPA within the same individuals, accounting for both domains, and associate these with musculoskeletal disorders. However, such designs are rare.

Most of the research on the physical activity paradox is based on self-reports, which are prone to inaccuracy. People appear to have difficulty estimating the time they spend in physical activities [21,22]. This may be particularly critical (and could even result in bias) when using self-reports at work and during leisure, considering that the physical activity likely differs. Thus, the understanding of the physical activity paradox can be deepened if data on OPA and LTPA are based on measurements using wearable sensors. A few studies have used accelerometry when studying exposures associated with outcomes such as wellbeing [23], all-cause sickness absence [24], sickness absence related to musculoskeletal disorders [25], and pain in the lower back and neck [26]. However, these studies did not assess the effects of both OPA and LTPA on musculoskeletal health, and only considered time spent in certain activities (e.g., sitting, standing and walking). The role of other exposures in OPA and LTPA, such as relative intensity, in determining musculoskeletal health are not yet well understood. 

The relative intensity of OPA has been shown to predict health outcomes, such as all-cause mortality [27] and imbalanced autonomic cardiac modulation [28], and may also provide insight into musculoskeletal health. The relative intensity of physical activity can be assessed through the percentage of heart rate reserve (%HRR), which is the percentage of a person’s total range in heart rate [29]. The %HRR depends on maximal and resting heart rate, which in turn depends on cardiorespiratory fitness. %HRR is a physiological metric, measuring the trade-off between a person’s physical activity and his/her capacity [30], and it allows for valid comparisons between participants. It has been shown that %HRR can be higher when the same physical activity is performed at work, compared to during leisure [31]. This is understandable, given that physical activity during work may include exposures that could influence %HRR beyond those found during leisure, for instance psychosocial demands [19,20,32,33]. Hence, investigating both OPA and LTPA in terms of %HRR in the same individuals, with musculoskeletal pain as the outcome, would provide further insight into the physical activity paradox.

Sleep disorders are also known to be associated with musculoskeletal health [34]. Since sleep, in addition to OPA and LTPA, is a part of the overall 24 h pattern of (in)activity, time spent sleeping should also be considered when studying the health effects of 24 h movement behavior. Compositional data analysis (CoDA) addresses in this context that time spent at work, in leisure, and in sleep are all inherently inter-related parts of a whole, i.e., 24 h [35,36]. CoDA views time spent in these various domains to be a correlated set of exposures and addresses this in a set of data processing procedures [35,36]. CoDA was introduced several decades ago [37] and has since then been extensively used in several fields of research, including geochemistry, economics, and nutrition. However, CoDA is quite new in (occupational) epidemiology [36]. Studies comparing data analyses with and without CoDA have shown that the two approaches lead to different results; the studies’ authors argue that the results are more correct when using CoDA [38,39]. Hence, applying CoDA to a data set reflecting 24 h recordings of OPA, LTPA, and sleep can result in new insights into determinants of musculoskeletal disorders, compared to analyses using standard methods [39].

We have collected information on 24 h heart rate data at baseline as well as musculoskeletal pain throughout a 2-year period among construction and healthcare workers [22,40,41]; both occupations have a relatively high OPA. The number of participants having complete 24 h data is relatively small, but the nature of the work and the 24 h recordings using wearable sensors make it well suited for exploring the physical activity paradox in terms of relative intensity of physical activity, using CoDA. Therefore, with this exploratory study, we aimed to investigate whether OPA and LTPA have different effects on musculoskeletal pain. Specifically, we explored associations of OPA and LTPA with musculoskeletal pain at three previously recommended thresholds of relative intensity, measured using the %HRR metric, and the extent to which OPA and LTPA differ in these associations. This study will contribute to a better understanding of the relationship between relative intensities of OPA and LTPA during work and leisure, and musculoskeletal pain. 

## 2. Materials and Methods

We recruited construction and healthcare workers to participate in a longitudinal cohort study of physical work demands and musculoskeletal pain [40]. Employees received information about the project at morning or lunch meetings; those unable to attend the meetings received the information from their manager/supervisors. Workers willing to participate filled in a questionnaire at baseline (594 construction and healthcare workers, response rate 51%). Of these employees, 371 gave their consent to participate in heart rate measurements. The number of measurements using wearable sensors were limited by the project’s capacity, and 138 participants were selected based on their job title to obtain a broad range of physical workloads found in each sector. Pregnant women and individuals diagnosed with a cardiovascular disease were excluded from this study. Baseline data collection started in the 1st quarter of 2014, with questionnaires and 24 h recordings of heart rate. Subsequently, individuals were approached every six months for two years with questionnaires; data collection ended in the 1st quarter of 2017.

Approval for this study was given by the Regional Committee for Medical and Health Research Ethics South East D in Norway (2014/138/REK). This study was conducted in accordance with the Helsinki Declaration. All participants gave written informed consent prior to the start of this study.

### 2.1. Independent Variable: Composition of Physical Activity during Work and Leisure

At baseline, we monitored heart rate using electrocardiography (ECG; Actiheart; Camntech, Cambridge, UK). The Actiheart has been found to reliably measure 24 h heart rate among physically active workers [16]. Standard ECG electrodes (Blue sensor VL-00-S/25 Ambu, Ballerup, Denmark) were attached at the apex of the sternum and on the left intercostal muscles at the level of the 6th and 7th costae [42]. Analogue signals were filtered (10 Hz–35 Hz) and sampled at a frequency of 128 Hz. The Actihearts were attached to the participants at the start of a working day and were continuously worn for 3–4 working days. Detailed information on this protocol can be found in previous publications [40,41]. 

The Actiheart recordings were processed using the custom-made Acti4 software (National Research Centre for the Working Environment, Copenhagen, Denmark and Federal Institute of Occupational Safety and Health, Berlin, Germany). Based on a short diary kept by the participants, the Actiheart data for each day were partitioned into periods before work, during work, after work, and during sleep. For inclusion in the present study, a 24 h recording had to contain data in the periods before and after work (combined into “leisure time”) as well as during work. 

Participants whose heart rate recordings contained more than 50% beat errors were excluded. The beat errors were defined as a heart rate <35 or >230 beats per minute, or an >15% difference in inter-beat time between two succeeding beat-to-beat periods. 

The relative intensity of physical activity was estimated for work and leisure separately by the percentage of heart rate reserve (%HRR), calculated using the ‘Karvonen method’, i.e., $\%HRR=\frac{HR-HRmin}{HRmax-HRmin}*100$. Maximal heart rate (HRmax) was estimated as 208−(0.7∗age) [43]. Minimum heart rate (HRmin) was defined as the minimum value of a running average of 10 beats during the time awake in all 3–4 assessment days.

Time spent above and below 20 %HRR, 30 %HRR, and 40 %HRR was then identified, both during work and during leisure. The threshold of 20 %HRR intends to discriminate sedentariness from a very light, but marked intensity of physical activity, including standing [31,44]. The 30 %HRR threshold defines activities that are at least light, such as moving and walking. This threshold was inspired by recommendations for the average physical activity to not exceed one-quarter to one-third of the relative aerobic strain throughout an 8 h working day [30,31,45]. The 40 %HRR threshold was selected, in accordance with the general recommendation suggested by the American College of Sports Medicine, to reflect whether exercise is performed at or above moderate intensity, i.e., a limit above which cardiorespiratory fitness is likely improved or maintained [46]. 

Sleep duration was included as part of the 24 h composition and calculated as the difference between 24 h and the sum of work and leisure time, as determined by the diary. Time spent sleeping, and time spent below and above the %HRR thresholds during both work and leisure, using each of the three thresholds, was averaged over the number of days recorded for each participant.

### 2.2. Dependent Variable: Overall Musculoskeletal Pain

The pain intensity question from the Pain Severity Index assessed the intensity of “Pain during the previous four weeks” on a 4-point scale (0 ‘no pain’, 1 ‘mild pain’, 2 ‘moderate pain’, 3 ‘severe pain’) [47]. Such categorical scales with verbal descriptors have been recommended when assessing pain intensity [48]. The pain intensity question for the neck/shoulder and upper limbs is similarly related to clinical signs as the Pain Severity Index in those two regions [47]. In the present study, the nine body regions from the ”Standardised Nordic questionnaire on musculoskeletal symptoms” were assessed, i.e., neck, left shoulder, right shoulder, elbows/wrists/hands, upper back, lower back, hips, knees, and ankles/feet [49]. The localization of the body regions was facilitated by a mannequin drawing from the Nordic questionnaire [49]. Pain intensity was reported at baseline and at a time point every six months during the 2-year follow-up. Pain scores were summed at every time point; thus, the pain score could range from 0 to 27 at any measurement occasion. 

### 2.3. Additional Descriptive Information 

At baseline, participants were asked to report their age (year of birth), gender, and smoking status (current smoker or quit smoking/never smoked). General health status (using the SF-36 single item, i.e., “How is your general health at present?” with five response alternatives ranging from excellent to poor) [50], as well as height and weight from which body mass index (BMI in kg/m^2^) was calculated, were reported at all follow-up time points. Perceived control of work pacing was also reported at every time point as the mean of four questions answered on a scale from 1 ‘very seldom or never’ to 5 ‘very often or always’ [51]. At baseline, participants were also asked to report their average weekly working hours, their perceived work ability (Work Ability Index single-item) [52], and seniority in their current occupation. Finally, at baseline, participants underwent a submaximal test on a cycle ergometer to estimate aerobic capacity (VO_2_max in ml/kg/min) [53] using the Åstrand nomogram [54] adjusted for age [55]. 

### 2.4. Data Analyses

Sample characteristics were summarized using descriptive statistics. All further analyses were conducted in R (version 4.1.1) (RStudio, Boston, MA, USA) using the ‘lme4′, ‘lmerTest’, and ‘compositions’ packages.

The exposure was summarized in terms of an exhaustive 5-part composition consisting of two parts for work (time spent below and above the %HRR threshold), two parts for leisure (time spent below and above the %HRR threshold), and one part for the time spent in sleep; these five parts added up to 100% of the 24 h day. The distributions of each of the parts of the 5-part composition were illustrated by standard cumulative distributions. In accordance with the CoDA approach, each exposure composition was then expressed as a set of four isometric log-ratio (ilr) coordinates [35,36]. The ilr coordinates were developed to address the relative time spent above and below the predefined %HRR thresholds at work and during leisure as follows: (1)$$ilr_{1}=\sqrt{\frac{4}{5}}\ln\left(\frac{\sqrt[{4}]{Work\;\%HRR\geq t\;*\;Work\;\%HRR<t\;*\;Leisure\;\%HRR\geq t\;*\;Leisure\;\%HRR<t}\;}{Sleep}\right)$$
(2)$$ilr_{2}=\ln\left(\frac{\sqrt[{2}]{Work\;\%HRR\geq t\;*\;Work\;\%HRR<t}}{\sqrt[{2}]{Leisure\;\%HRR\geq t\;*\;Leisure\;\%HRR<t}}\right)$$
(3)$$ilr_{3}=\sqrt{\frac{1}{2}}\ln\left(\frac{Work\;\%HRR\geq t}{Work\;\%HRR<t}\right)$$
(4)$$ilr_{4}=\sqrt{\frac{1}{2}}\ln\left(\frac{Leisure\;\%HRR\geq t}{Leisure\;\%HRR<t}\right)$$
where *t* is the %HRR threshold, i.e., 20%, 30%, or 40%. After *ilr* transformation, data were analyzed using standard statistical methods [56,57].

Setting up the *ilr*s as described in the equations is a novel way of viewing the distribution of time spent throughout the 24 h day, which differs from previously used *ilr* sets [38,56,58,59]. Previously, one part of the composition was inspected relative to all other parts, and changes in that one part were typically interpreted relative to time spent in all other parts. In our approach, the relative proportion of time spent above the predefined threshold at work (*ilr*_3_) and during leisure (*ilr*_4_) can be directly interpreted, relative to time spent below these thresholds in these same domains. This novel approach to the ilr structure gives direct insight into how physical activity at work, as well as physical activity during leisure, is associated with an outcome. The two remaining ilrs, i.e., *ilr*_1_ and *ilr*_2_, are included to give a complete expression of the distribution of time spent during the day, as required in the CoDA approach, even though they do not explicitly answer the research question in the present study.

The outcome, i.e., overall musculoskeletal pain, was analyzed under the assumption that ratings could be treated as values on a continuous scale from 0 to 27. Associations between the exposure compositions for each of the three physical activity thresholds (≥20%, ≥30%, and ≥40 %HRR) and overall musculoskeletal pain during the 2-year follow-up were analyzed using linear mixed models with random intercept for participants. Three separate models were developed, one for each physical activity threshold. Each model included the four ilr coordinates, and time during follow-up as a continuous variable from 0 to 4 (with the value ‘0′ corresponding to baseline) to account for the repeated pain assessments. Visually inspected quantile–quantile plots indicated that normality assumptions of residuals after regression were met. 

The extent to which VO_2_max, control of work pacing at baseline, BMI at baseline, and occupational sector modified the associations between the physical activity compositions and overall musculoskeletal pain was assessed by stratified analyses. VO_2_max and control of work pacing were dichotomized at the median (34.8 mL/min/kg and 2.75, respectively), BMI was dichotomized into normal (<25 kg/m^2^) and overweight/obese (≥25 kg/m^2^), and sector into construction and healthcare. 

### 2.5. Visualization of the Regression Analyses

Results of the regression analyses were illustrated in ‘heat maps’ for each %HRR threshold. First, we calculated the arithmetic mean time spent in work, leisure, and sleep in the sample. Second, we solved the regression equations for 100 predefined steps in percentage time spent above each %HRR threshold during work, while keeping the total time at work, the total time in leisure, and the percentage of time spent above the %HRR threshold during leisure constant. Then, this process was repeated for 100 predefined steps in percentage time spent above the %HRR threshold during leisure, i.e., 10,000 simulations in total, all of which had constant total times at work and in leisure, corresponding to the observed mean in the sample. The 100 predefined steps were selected to represent the full range of percentages of time spent above each %HRR threshold in the source data.

## 3. Results

Twelve of the 138 workers who were selected to participate with heart rate assessments were unable to do so (Figure 1). Twenty-four of the remaining 126 participants wore heart rate monitors that did not function or had >50% beat errors. A further 30 participants missed data for some of the work or leisure time. The final study sample consisted of 72 participants having complete recordings for at least one day (work + leisure). Forty-eight participants had recordings for one complete day and 24 participants had recordings for two complete days.

### 3.1. Sample Characteristics

The construction and healthcare sectors were equally represented in the sample; a slightly larger proportion of participants were male (Table 1). The average age of participants was 42.5 years, and they had an average of 17 years of work experience in their current occupation. BMI was on average 25.6 kg/m^2^; 51% were overweight/obese, i.e., they had a BMI > 25 kg/m^2^. VO_2_max was on average 35.6 mL/kg/min, with 30.9 mL/min/kg (sd 7.8) for women and 38.7 mL/kg/min (sd 11.1) for men, which appears quite low for both genders [60]. Just under one-third of the sample were current smokers, the participants generally perceived their work ability to be high, and the majority reported their health to be good to excellent. Overall pain at baseline was, on average, 5.6 on the 0–27 scale (sd 5.0). The participants in the present study (*n* = 72) did not differ substantially in any of these properties from those who only answered questionnaires at baseline (*n* = 522).

### 3.2. Exposure Compositions

The average working day lasted 7 h 51 min (sd 1 h 57 min), and the average leisure time was 9 h 34 min (sd 2 h 16 min). The participants slept on average for 6 h 34 min (sd 1 h 20 min). During work and during leisure, the participants spent on average more than 5 h (21–23% of the day) in activities requiring a heart rate ≥20 %HRR; more than 2 h (9–10% of the day) in activities requiring ≥30 %HRR, and approximately 45 min (3% of the day) in activities requiring ≥40 %HRR (Table 2, Figure 2). 

### 3.3. Physical Activity in Relation to Overall Musculoskeletal Pain

Pain changed little over time (e.g., in the model for ≥20 %HRR ß 0.01, 95% CI (−0.22–0.23); Table 3). Although none of the associations for physical activity and overall musculoskeletal pain were statistically significant, this exploratory study still reveals some interesting trends for further research regarding the direction and size of the beta-coefficients. First, the results indicate that more time spent awake, relative to asleep, was associated with lower overall musculoskeletal pain scores (indicated by the negative beta-coefficients for *ilr*_1_ in each model, e.g., in the model for ≥20 %HRR ß −1.18, 95% CI (−5.08–2.72). Second, longer time at work (vs leisure time) was also associated with lower overall pain scores (indicated by the negative beta-coefficients for *ilr*_2_, e.g., for ≥20 %HRR ß −0.53, 95% CI (−2.43–1.38)). Further, the results indicate consistently that irrespective of the threshold level (20 %HRR, 30 %HRR, or 40 %HRR), that participants spending more time in physical activity above the thresholds at work (relative to below the thresholds) had higher pain scores. The fact that associations for work (expressed by *irl*_3_) were almost equally strong for the three %HRR thresholds indicates that high intensities of OPA, i.e., ≥40 %HRR, are decisive for the pain outcome. In contrast, the more time they spent in activity during leisure above the three thresholds (relative to below the thresholds; expressed in *irl*_4_), the less pain participants perceived. However, the effect of LTPA faded away as the intensity threshold increased: the strongest association was found for the proportion of time spent ≥20 %HRR, relative to <20 %HRR. This indicates that time in ‘any’ physical activity, i.e., time even at intensities ≥20 %HRR, has a bearing on the positive effect on pain development. 

Figure 3a–c illustrates the resolved regression equations and represents an alternative to the isotemporal substitutions used in many studies applying CoDA. The figures suggest that, depending on the time spent in LTPA, the pain score was approximately 6 if 40–100% of the working time was spent at intensities ≥20 %HRR (Figure 3a), or if 20–70% of the working time was spent ≥30 %HRR (Figure 3b), or if 10–30% of the working time was spent ≥40 %HRR (Figure 3c). Figure 3a shows that more time in LTPA ≥20 %HRR was associated with lower risk of musculoskeletal pain, also when time in OPA ≥20 %HRR increased. We found a weaker but similar trend for LTPA ≥30 %HRR (Figure 3b), while the influence of time spent ≥40 %HRR during leisure seemed small (Figure 3c). 

Appendix A show that aerobic capacity (VO_2_max), BMI, control of work pacing, and occupational sector modified the associations between the physical activity compositions and overall musculoskeletal pain. Most notable, participants with higher aerobic capacity (VO_2_max above 34.8 mL/min/kg) had more benefit from spending time in higher LTPA than those with lower capacity. Additionally, those with a lower BMI (<25 kg/m^2^) had lower pain scores when more time was spent in more intense activity during work as well as during leisure, compared to those with a higher BMI (>25 kg/m^2^), irrespective of the physical activity threshold. Additionally, having more control of work pacing (≥2.75) tended to be associated with higher pain scores than having less control of work pacing. 

## 4. Discussion

Even though the results of this exploratory study did not show any statistically significant associations between time spent above predefined physical activity thresholds during work or leisure and overall musculoskeletal pain, some interesting findings call for more research. In general, participants spending more time at work above the physical activity thresholds (relative to below the thresholds) had higher pain scores, while participants with relatively more time during leisure above the activity thresholds had lower pain scores. Additionally, this effect of OPA to increase pain was weaker for participants with higher aerobic capacity or lower BMI, while more LTPA decreased pain for these individuals.

### 4.1. Strengths and Weaknesses

This exploratory study had several strengths. First, the relative intensity of physical activity was assessed using wearable sensors measuring heart rate over 1–2 days. Specifically in a work setting, such measurements are more accurate and less prone to bias than self-reported measures of relative intensity, for instance using Borg ratings [61,62]. Second, the novel set of *ilr*s developed for this study enabled us to study the physical activity paradox as a within-subject phenomenon, by comparing time spent in physical activity during work with time spent in physical activity during leisure. Third, we used a novel approach that facilitated the interpretation of the regression equations by solving the equations for an average worker and depicting the results in a heat map. In this way, the sizes of the beta-coefficients for the *ilr*s, that cannot directly be interpreted, were transformed into meaningful effects so that time spent in physical activity during work and leisure could be related to overall musculoskeletal pain. Last, the repeated assessments of pain over the 2-year follow-up provided more reliable estimates of average pain than single measurements and, thus, analyses with better power. 

One major weakness of our study is related to data quality. Data from several participants had to be deleted, which decreased the power of our analyses and made it difficult to identify statistically significant associations. However, when comparing the properties of our sample (*n* = 72) with those who answered questionnaires only at baseline (*n* = 522), we found no substantial differences. This suggests that had we had a larger sample of heart rate data, our findings might have been more conclusive, thus providing better support for the explorative findings that the physical activity paradox may apply to cardiovascular intensity and musculoskeletal pain. Further, our ability to assess associations between physical activity and pain would have been improved, had we assessed heart rate at several time points throughout the 2-year follow-up. The need to have repeated exposure measurements is a general reminder to all studies of occupational exposures. Additionally, the participants in our study were a selected group of physically active workers with a good self-reported work ability and many years of work experience, which may indicate a healthy worker survival effect. Interestingly, the sample also had quite a low VO_2_max, which may raise the question whether a ‘high’ VO_2_max is needed to easily perform physically demanding work tasks amongst experienced workers.

### 4.2. Interpretation of the Findings

The results from our study suggest that the ‘physical activity paradox’ applies even to associations between cardiovascular work intensity and musculoskeletal pain, complementing previous studies of cardiovascular health [2,3,4], cancer [5], and all-cause mortality [6]. Even though participants spent similar amounts of time in physical activity at work and during leisure (cf. Table 2 and Figure 3a–c), the associations with pain differed between OPA and LTPA for all intensity thresholds, suggesting that time spent in OPA may have a detrimental impact on overall pain, while LTPA may have a beneficial impact. 

Our results further suggest that the importance of the intensity of physical activity differs between work and leisure. For work, a high intensity, ≥40 %HRR, was associated with pain, while for leisure, high intensities did not contribute much. The associations between LTPA and overall musculoskeletal pain were strongest for time spent ≥20 %HRR (Figure 3), which may indicate that some LTPA may help alleviate overall musculoskeletal pain, especially when, at the same time, spending less time in OPA ≥20 %HRR. However, LTPA at a higher intensity, i.e., ≥40 %HRR, may have contributed little to reduce pain, especially when also spending time in higher-intensity OPA, i.e., ≥40 %HRR. The result that a longer duration in OPA, in terms of relative intensity, may be associated with pain in several bodily regions corroborates previous studies, showing that physical work demands measured with other metrics are associated with multi-site pain [63,64,65].

Additionally, our results suggest that lower physical fitness may increase the risk of musculoskeletal pain, likely due to OPA being performed at a relatively high intensity. This agrees with results in Holtermann et al. [66], reporting that lower physical fitness increased the risk of cardiovascular mortality among workers with high physical demands. Interestingly, findings reported by Ketels et al. [67] suggest that more OPA does not improve physical fitness, while more LTPA does. Additionally, our results indicate that lower BMI may be protective in the association between physical activity and overall pain, irrespective of the intensity level or whether the activity is performed during work or leisure. This is consistent with a considerable body of knowledge, showing that overweight and obese individuals are more likely to experience pain [68,69,70]. 

We may speculate what explains the apparently differential associations between OPA and LTPA with musculoskeletal pain, considering that it appears physiologically implausible that similar intensity levels maintained for similar durations would have a differential impact on health, depending on whether they are performed at work or during leisure. Holtermann, Krause, van der Beek and Straker [17] argued that OPA may be of too low intensity or too long duration, compared to LTPA, to maintain or improve cardiovascular health. Our study, while on musculoskeletal pain, does not support this theory, considering that, on average, our study participants spent similar total amounts of time at the different activity intensities during work and leisure (Table 2), while the impact of OPA seemed to be detrimental and the impact of LTPA seemed to be beneficial. What may, however, explain the differential effect of OPA and LTPA on musculoskeletal health is the different temporal structure of activities performed during work and leisure. OPA is characterized by prolonged static postures, manual handling, and heavy lifting, while LTPA is characterized by more sporadic, whole-body dynamic movements such as running and cycling. These different patterns of activities may have different impacts on musculoskeletal health, and be of major relevance, in addition to the total time at different intensities, as suggested by basic and applied work physiology [71,72].

The likely importance of the temporal pattern also emphasizes the relevance of OPA often being performed without sufficient time to recover, while LTPA is performed under controlled conditions that do provide sufficient recovery [17,73]. Insufficient time to recover during work or between workdays may not give tissues an opportunity to heal and may elevate levels of inflammation in the body; both factors may explain higher perceived pain levels [74,75].

Interestingly, our study’s findings contrast with previous studies regarding the associations with pain of some of the participant characteristics. First, Holtermann, Krause, van der Beek and Straker [17] propose that less favorable psychosocial conditions, such as low worker control, may increase the effect sizes in the association between OPA and cardiovascular health. Our study suggests the opposite, i.e., that less worker control reduced the risk of musculoskeletal pain. Second, previous studies show that shorter sleep duration is related to increased risk of pain [76,77], while our findings suggest that the longer participants were awake (relative to sleep duration), the lower their pain scores were. Third, our results suggesting that longer work time (relative to leisure time) may be associated with more pain contrasts previous findings showing that longer working hours is related to increased pain [78]. One of the explanations for the contrasting findings could lie in the different analysis methods, i.e., previous studies are based on standard analyses using absolute durations, while the results in our present study are based on CoDA that uses relative durations. 

### 4.3. Future Directions

More studies of larger samples with better statistical power are warranted to support or contradict the trends of the current exploratory study. Future studies may draw inspiration from the current study when investigating the differential health effects of total time at work and leisure by using the same approach, i.e., the same set of *ilr*s. To investigate the differential effects further, repeated exposure assessments over multiple days at a time would be beneficial. Additionally, future studies aiming to understand the physical activity paradox would benefit from studying effect modification by stratified analyses, as performed in the current study, rather than solely adjusting for confounders. Last, we assessed only total time spent at work and in leisure at different intensities (relative to lower intensities) of physical activity, while we encourage future studies to include metrics quantifying temporal patterns of physical activity, for instance time spent in periods (‘bouts’) of different intensity. This will likely provide more insight into why the paradox exists, and even if there is, indeed, a paradox.

## 5. Conclusions

The current exploratory study suggests that the physical activity health paradox may also apply to musculoskeletal pain. Our results suggest that workers spending relatively more time in OPA have higher pain, while workers with relatively more time in LTPA have less pain. However, none of the associations were statistically significant. The impact on pain of OPA was less pronounced for participants with higher aerobic capacity or lower BMI, while LTPA was beneficial for these individuals. Future studies of other occupational groups, and with larger sample sizes, are needed to verify or dismiss the trends found in the current study. We also encourage studies to investigate the temporal patterns of activity, including even psychosocial factors, to further explore the physical activity health paradox. 

## Figures and Tables

**Figure 1 ijerph-19-02751-f001:**
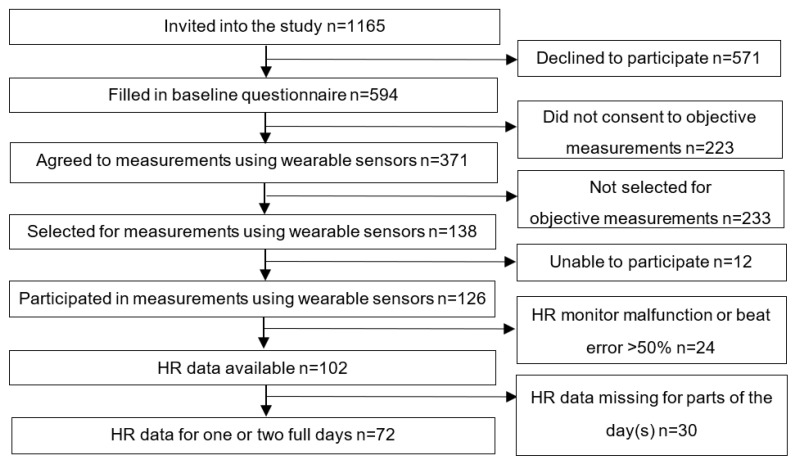
Flow diagram showing the selection procedure of participants for whom we had heart rate (HR) data for both work and leisure for at least one full 24 h day.

**Figure 2 ijerph-19-02751-f002:**
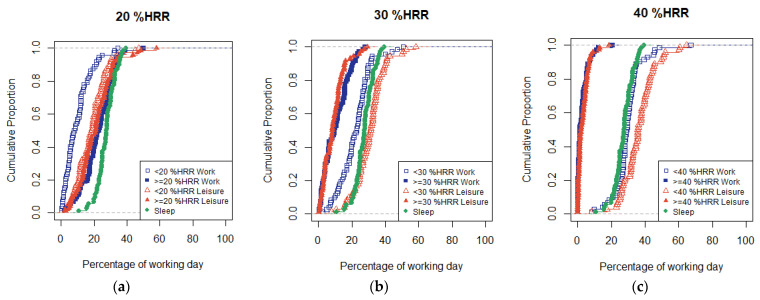
Cumulative distribution functions for the participants’ (*n* = 72) percentage of time in sleep and in activity below and above (**a**) 20 %HRR, (**b**) 30 %HRR, and (**c**) 40 %HRR during work and leisure.

**Figure 3 ijerph-19-02751-f003:**
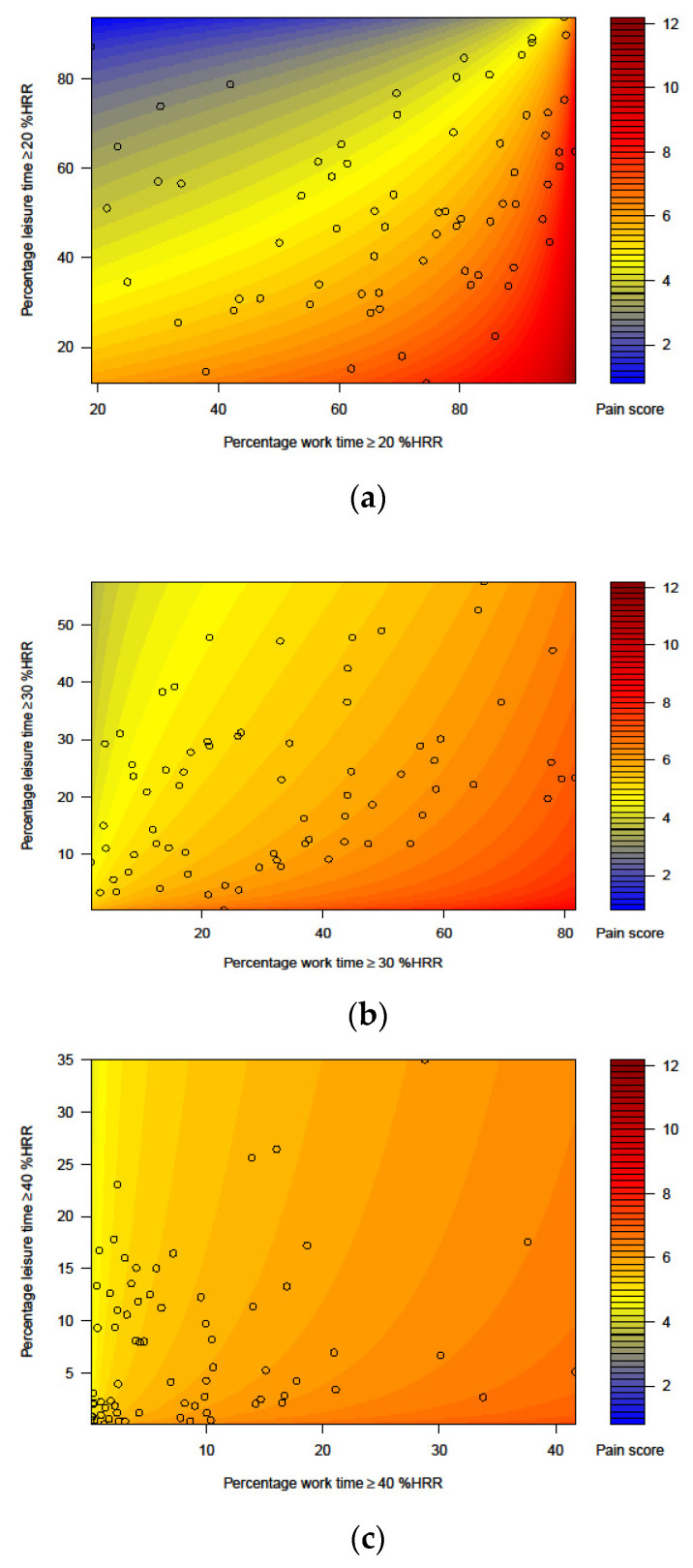
Heat maps illustrating the predicted average overall musculoskeletal pain score (scale from 0 to 27) over the 2-year follow-up at different baseline percentages of time (**a**) ≥20 %HRR, (**b**) ≥30 %HRR, and (**c**) ≥40 %HRR at work and during leisure. In each panel, the black circles show individual, observed compositions of time at work and in leisure. The figures illustrate the pain score according to the regression model only within the range of compositions represented in the sample (*n* = 72).

**Table 1 ijerph-19-02751-t001:** Participant characteristics at baseline.

	Total *(n = 72)*			
	*n*	(%)	mean	(sd)
**Age (years)**			42.5	(11.1)
**Seniority (years)**			17	(11.2)
**Work hours (per week)**			36.7	(4.4)
**Gender**				
Male	43	(60)		
Female	29	(40)		
**Sector**				
Construction	35	(49)		
Healthcare	37	(51)		
**BMI (kg/m^2^)**			25.6	(3.5)
**BMI**				
Normal (<25 kg/m^2^)	35	(49)		
Overweight/obese (≥25 kg/m^2^)	37	(51)		
**VO_2_max (ml/min/kg)**			35.6	(10.5)
**Smoking**				
Yes (current smoker)	20	(28)		
No (never or quit smoking)	52	(72)		
**Work ability (scale 0–10)**			8.9	(1.4)
**Control of work pacing at baseline (score 1–5)**			2.8	(0.8)
**General health**				
Poor or fair	11	(15)		
Good	28	(39)		
Very good or excellent	33	(46)		
**Overall musculoskeletal pain at baseline (0–27)**			5.6	(5.0)

The main variable is in bold.

**Table 2 ijerph-19-02751-t002:** Averages of time and percentage of time during a 24 h day spent in the physical activity categories and in sleep.

	Mean Time in a Day (Minutes)		Proportion of the Day (% Time)	
	*n* = 72		*n* = 72	
**20 %HRR**	**mean**	**(sd)**	**mean**	**(sd)**
≥20 %HRR Work	327	(132)	22.7	(9.1)
<20 %HRR Work	145	(115)	10.0	(8.0)
≥20 %HRR Leisure	305	(152)	21.2	(10.5)
<20 %HRR Leisure	269	(133)	18.7	(9.2)
Sleep	394	(80)	27.4	(5.6)
Total	1440		100	
**30 %HRR**				
≥30 %HRR Work	154	(107)	10.6	(7.4)
<30 %HRR Work	318	(133)	22.1	(9.3)
≥30 %HRR Leisure	131	(93)	9.1	(6.5)
<30 %HRR Leisure	443	(130)	30.8	(9.0)
Sleep	394	(80)	27.4	(5.6)
Total	1440		100	
**40 %HRR**				
≥40 %HRR Work	43	(52)	3.0	(3.6)
<40 %HRR Work	429	(120)	29.7	(8.3)
≥40 %HRR Leisure	47	(50)	3.3	(3.4)
<40 %HRR Leisure	527	(138)	36.6	(9.6)
Sleep	394	(80)	27.4	(5.6)
Total	1440		100	

The main variable is in bold.

**Table 3 ijerph-19-02751-t003:** Models describing the association between time spent above 20 %HRR, 30 %HRR, and 40 %HRR and overall musculoskeletal pain (*n* = 72).

%HRR threshold	ß	(95% CI)
**20 %HRR**		
Intercept	4.06	(1.67–6.45)
Time	0.01	(−0.22–0.23)
Awake/Sleep *(ilr*_1_)	−1.18	(−5.08–2.72)
Work/Leisure (*ilr*_2_)	−0.53	(−2.43–1.38)
≥20 %HRR Work/<20 %HRR Work (*ilr*_3_*)*	1.01	(−0.35–2.36)
≥20 %HRR Leisure/<20 %HRR Leisure (*ilr*_4_)	−1.56	(−3.29–0.17)
**30 %HRR**		
Intercept	5.22	(2.33–8.10)
Time	0.01	(−0.22–0.23)
Awake/Sleep (*ilr*_1_)	−0.56	(−4.88–3.75)
Work/Leisure (*ilr*_2_)	−0.27	(−2.26–1.71)
≥30 %HRR Work/<30 %HRR Work (*ilr*_3_)	0.92	(−0.63–2.48)
≥30 %HRR Leisure/<30 %HRR Leisure (*ilr*_4_)	−0.57	(−2.59–1.45)
**40 %HRR**		
Intercept	6.08	(2.77–9.36)
Time	0.01	(−0.22–0.23)
Awake/Sleep (*ilr*_1_)	−1.02	(−5.60–3.54)
Work/Leisure (*ilr*_2_)	−0.32	(−2.47–1.84)
≥40 %HRR Work/<40 %HRR Work (*ilr*_3_)	0.98	(−1.28–3.24)
≥40 %HRR Leisure/<40 %HRR Leisure (*ilr*_4_)	−0.15	(−2.23–1.94)

The main variable is in bold.

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
