# Peer review of "An Exploratory Study on the Physical Activity Health Paradox—Musculoskeletal Pain and Cardiovascular Load during Work and Leisure in Construction and Healthcare Workers"

_ijerph, 2022, doi:10.3390/ijerph19052751_

Round 1
Reviewer 1 Report
This is a very good assessment of the the physical activity and pain levels of adults over time while they are in recreation and the workplace. The data are well presented and explained. Additionally, the relevance of the presented data to the current literature is concisely written. Overall. this manuscript does well in presenting the objectives, explaining the methodology, reporting the data, and interpreting the data interwoven with the currently understood research.
Author Response
We would like to thank the reviewer for taking the time to review our manuscript; we appreciate your positive review.
Reviewer 2 Report
Dear Authors,
Your paper is very well written and presents interesting work for a large audience in human resource management and health care experts. There are very few issues in the paper, I only have two:
- The lack of significant results is not satisfying and could be improved - in my opinion - if you depart from the idea that your research is of an exploratory/discovery type. Your argument in lines 95 – 99 does not convince me of your data.
Below I exemplify a hypothesis and a subsequent statistical test to show that part of your research can be proved to be significant. - Your novel ilr estimator in the compositional data analysis needs a slight improvement. The main reason is that it is multiplicative. This latter property has the disadvantage that it ‘multiplies away’ effects of the experiment. This latter might have caused A.
Below I discuss this in detail.
Conclusion: your exploratory assumption is redundant. By stating a research hypothesis I can prove that the research has a significant result for discriminating between Occupational Physical Activity (OPA) and Leisure Time Physical Activity (LTPA)) at the lower intensities of pain. Not the higher intensities. In the addended file, I give computational proof, with the use of Maple software from MapleSoft Inc. Canada.
It was an honor to review such good and interesting work.
Details, suggestions, and discussions:
- About A.: your paper reports exploratory work and consequently, you use an exploratory statistical analysis method. This exploratory/discovery character inhibits you to find significant outcomes of your substantial and impressive experiment. In contrast to this, I invoke a Null hypothesis, - by a non-exploratory statistical test – that partly yields significance.
I take a Null hypothesis from your lines 61, 62, and from your lines 397 – 398, 437 – 441.
To remedy this I propose to invoke from statistics the F-distribution for sustaining detailed analysis, without this loss of information. It is explained in https://en.m.wikipedia.org/wiki/Fisher%27s_exact_test, with more details in for instance Hogg & Craig 1. - Your novel ilr structure in compositional data analysis needs a slight improvement. The main reason is that the current (novel) version is ( in the estimators ilr1 and ilr2 ) designed to be multiplicative. This latter property has the disadvantage that it ‘multiplies away’ effects of the experiment. On line 368 on page 13 you complain about the metric.
In analogy to the property of summative estimators: known as 'Regression to the mean', your multiplication of fractions in ilr1 has the property of 'Iterating to zero'.
This property explains why your outcomes do lose information. Note the very different and revealing results in the heat maps (a), (b), (c) in Figure 3.
To exemplify why I am saying this, let me denote
p1 := Work %HRR t
p2 := Work %HRR < t
p3 := Leisure %HRR t
p4 := Leisure %HRR < t
p1, p2, p3, p4 all are in the interval between 0 and 1, so the product p1 • p2 • p3 • p4 is less than any of its individual terms.
Concluding: p1 • p2 • p3 • p4 < minimum{p1, p2, p3, p4}
This multiplicative property of the estimator annihilates information from the data. This might be unwanted and seems to be wrong in your purpose of the design. - Page 4, line 155: Shouldn’t the minimum value of the heartbeats be taken at rest?
- Page 4, line 160: ‘to’ should read: ‘too.
- Page 4, line 161: ‘not exceed’ should read: ‘not exceeding’.
- Page 4, line 176: ‘and every six months’ is unclear for me. Is it ‘and at a time point every six months'?
- Page 4, line 182: ‘using the SF-36 single item’ is unclear, it assumes reading the paper [46]. Could you explain this?
- Page 4, line 189: ‘the the’ is redundant.
- Page 4, line 197: I suggest to insert after ‘sleep;’: the duration intervals of these.
- Page 4, line 198: where is the 5-part composition illustrated?
- Page 5, lines 232 – 241: the description of the method of visualization by heat maps could be better placed in a separate section (in the methods chapter) about descriptive visualization. Particularly the use of the word ‘simulation’ should be avoided in a section about data-analysis.
- Page 8, in the capture of Figure 2 you say ‘probability/distributions’, while you display - I think – proportion scores. The distributions are assumptions to be tested. To me, this is confusing, as if you replace assumptions with data outputs.
- Page 11, figure 3 shows why I argue about replacing your multiplicative version of your metrics ilr1 and ilr2 by a different version: the heat maps invoke a rich view on deeper results from your data. I think I scratched only the surface with my suggestion for a (profile) Chi square test.
- Pages 11 – 19: incomplete references with missing page numbers are abundant. For instance: references , 64, 72, 73, 75.
- Page 13, line 366, ‘solving’ should read: ‘fitting’.
- Page 19, lines 640 – 642: Maybe reference 47 needs scrutiny? It seems incomplete.
- Page 19, line 646: inconsistent lay-out: year 2003 not written bold. Maybe reference 49 needs scrutiny?
- Page 19, lines 665: Maybe reference 57 needs scrutiny? It seems incomplete.
Reference
- Hogg R V, Craig AT. Introduction to Mathematical Statistics, 7th ed. The Macmillan Company, London, 1972. 1–415 pages.
Author Response
General response
Thank you for your thorough review of our manuscript and thank you for your compliments. In our point-by-point responses below we address your comments and concerns.
Comment 1
The lack of significant results is not satisfying and could be improved - in my opinion - if you depart from the idea that your research is of an exploratory/discovery type. Your argument in lines 95 – 99 does not convince me of your data.
Below I exemplify a hypothesis and a subsequent statistical test to show that part of your research can be proved to be significant.
Response to comment 1:
We get the impression that the reviewer finds it important to detect statistically significant associations between the exposure and outcome in our study. However, we think that finding statistical significance should not be a goal in itself. Rather, following the lively discussion surrounding the often wrong interpretation of statistical significance (e.g. see the special issue in the American Statistician from 2019 ”Statistical inference in the 21st Century: a World beyond p<0.05” [1]), we chose to interpret the meaningfulness of effect sizes, including confidence intervals, in light of the sample size and variability in the study [2].
We would like to highlight that we mention in the last paragraph of the introduction that our sample is small and that we used a relatively large number of statistical analyses in order to address our question of whether the physical activity paradox in relation to pain may exists for different threshold-levels of physical activity. These are the two main reasons why we chose to call our study ‘exploratory’, and to refrain from testing any formal hypothesis. We did this because we did not want to risk arriving at any wrong conclusions. In addition to our conclusions concerning OPA and LTPA, our study can be used as an exploration into the use of CoDA in occupational health sciences, including the novel ilr-structure, as we suggest in the discussion, and also into our novel approach of interpreting the results using heat maps. However, as the reviewer has correctly observed, we have operated with an underlying theory, that the physical activity paradox could be extended to musculoskeletal pain.
However, the data processing technique applied, i.e. Compositional Data Analysis (CoDA), is not ‘exploratory’ per se. The basic principles and rationale of CoDA were introduced by John Aitchison several decades ago [3], as a response to the need to correctly analyze parts of a whole which are mutually dependent and multicollinear. Since the introduction of CoDA, it has been used in various fields of research including e.g., environmental sciences, economics, and nutrition [4-6]. However, CoDA is relatively new in (occupational) epidemiology and has mostly been used in studies on the time spent in various physical activities and their association with, for example, adiposity and cardio-metabolic health [7-9], and sickness absence [10,11].
We have adjusted the text in lines 95-99 to clarify our choice of CoDA and to hopefully guide our future readers even better:
Introduction, paragraph 6: ”Compositional data analysis (CoDA) addresses in this context that time spent at work, in leisure, and in sleep are all inherently inter-related parts of a whole, i.e. 24h [35,36]. CoDA views time spent in these various domains to be a correlated set of exposures and addresses this in a set of data processing procedures [35,36]. CoDA was introduced several decades ago [37], and has since then been extensively used in several fields of research, including geochemistry, economics, and nutrition. However, CoDA is quite new in (occupational) epidemiology [36]. Studies comparing data analyses with and without CoDA have shown that the two approaches lead to different results; the studies’ authors argue that the results are more correct when using CoDA [38,39]. Hence, applying CoDA to a data set reflecting 24h recordings of OPA, LTPA, and sleep can result in new insights into determinants of musculoskeletal disorders, compared to analyses using standard methods that do not use CoDA [39].”
Comment 2
Your novel ilr estimator in the compositional data analysis needs a slight improvement. The main reason is that it is multiplicative. This latter property has the disadvantage that it ‘multiplies away’ effects of the experiment. This latter might have caused A.
Below I discuss this in detail.
Response comment 2:
We appreciate reading your the ideas. However, when using CoDA, we must follow the mandatory principles included there. This implies a set of procedures involving multiplicative elements, since they are essentially based on log-transformed geometric averages and ratios of components in data [3,12,13]. Hence, we cannot change the ilr-coordinates in a way that violates the inherent build of them. We agree with the reviewer that the use of CoDA implies several consequences, which are not easy to handle, and not yet understood to their full extent, and we look forward to continued discussion about CoDA. See also our more in-depth responses below under responses 4 and 5.
Comment 3 - Conclusion:
Your exploratory assumption is redundant. By stating a research hypothesis I can prove that the research has a significant result for discriminating between Occupational Physical Activity (OPA) and Leisure Time Physical Activity (LTPA)) at the lower intensities of pain. Not the higher intensities. In the addended file, I give computational proof, with the use of Maple software from MapleSoft Inc. Canada.
Response 3
We have provided a detailed response to your comment about our exploratory assumptions and the use of CoDA to process our data under Reponses 1 and 4.
Details, suggestions, and discussions:
Comment 4
About A.: your paper reports exploratory work and consequently, you use an exploratory statistical analysis method. This exploratory/discovery character inhibits you to find significant outcomes of your substantial and impressive experiment. In contrast to this, I invoke a Null hypothesis, - by a non-exploratory statistical test – that partly yields significance.
I take a Null hypothesis from your lines 61, 62, and from your lines 397 – 398, 437 – 441.
To remedy this I propose to invoke from statistics the F-distribution for sustaining detailed analysis, without this loss of information. It is explained in https://en.m.wikipedia.org/wiki/Fisher%27s_exact_test, with more details in for instance Hogg & Craig 1.
Response 4
Comment 5
Your novel ilr structure in compositional data analysis needs a slight improvement. The main reason is that the current (novel) version is ( in the estimators ilr1 and ilr2 ) designed to be multiplicative. This latter property has the disadvantage that it ‘multiplies away’ effects of the experiment. On line 368 on page 13 you complain about the metric.
In analogy to the property of summative estimators: known as 'Regression to the mean', your multiplication of fractions in ilr1 has the property of 'Iterating to zero'.
This property explains why your outcomes do lose information. Note the very different and revealing results in the heat maps (a), (b), (c) in Figure 3.
To exemplify why I am saying this, let me denote
p1 := Work %HRR t
p2 := Work %HRR < t
p3 := Leisure %HRR t
p4 := Leisure %HRR < t
p1, p2, p3, p4 all are in the interval between 0 and 1, so the product p1 • p2 • p3 • p4 is less than any of its individual terms.
Concluding: p1 • p2 • p3 • p4 < minimum{p1, p2, p3, p4}
This multiplicative property of the estimator annihilates information from the data. This might be unwanted and seems to be wrong in your purpose of the design.
Response 5
Thank you once again for your suggestions. However, when using CoDA, we cannot deviate from its principles, which implies that we cannot change the ilr-coordinates in such a way that they will violate CoDA principles. The product of the term p1 • p2 • p3 • p4 in ilr1 indeed returns a number that is less than any of its individual terms; however, we want to remind the reviewer that when we take the fourth root of the product, a value is returned that is not close to zero. Thus, the data do not ’iterate to zero’ or multiply away the effects of the study.
The ilrs are an integrated and interrelated element in CoDA, and we followed the CoDA principles to process the data, as also explained above. The novelty of the ilr estimator lies in the way we partitioned the data, which is different from previous studies using CoDA (lines in resubmitted document 229-231): ”Previously, one part of the composition has been inspected relative to all other parts, and changes in that one part was typically interpreted relative to time spent in all other parts.” We have partitioned the data in such a way that we can directly relate higher to lower OPA, as well as directly relate higher to lower LTPA, rather than relating higher OPA or LTPA to the remaining domains of the 24h period, as previously has been done.
Comment 6
Page 4, line 155: Shouldn’t the minimum value of the heartbeats be taken at rest?
Response 6
Indeed, the minimum heartbeat during rest is what is defined as HRmin in the Karvonen method. We did not assess heart rate during rest. Instead, we have 24h recordings of heart rate, for which it is highly likely that participants at some point in time were, indeed, at rest. By taking the minimum heart rate throughout 3-4 days, we assumed that we used the lowest possible heart rate as the minimum heartbeat during rest.
Comment 7
Page 4, line 160: ‘to’ should read: ‘too.
Response 7
We have slightly reworded the sentence for clarity:
Materials and Methods, Independent variable: composition of physical activity during work and leisure, paragraph 5: “This threshold was inspired by recommendations for the average physical activity not to exceed a quarter to one third…”.
Comment 8
Page 4, line 161: ‘not exceed’ should read: ‘not exceeding’.
Response 8
Please see our response to the preceding comment.
Comment 9
Page 4, line 176: ‘and every six months’ is unclear for me. Is it ‘and at a time point every six months'?
Response 9
Thank you. We have followed the reviewer’s suggestion and clarified this part of the text to: ’and at a time point every six months’.
Comment 10
Page 4, line 182: ‘using the SF-36 single item’ is unclear, it assumes reading the paper [46]. Could you explain this?
Response 10
We have added the single item of the SF-36 to the text:
Materials and Methods, Additional descriptive information: “General health status (using the SF-36 single item, i.e. “How is your general health at present?” with five response alternatives ranging from excellent to poor) [46])…”
Comment 11
Page 4, line 189: ‘the the’ is redundant.
Response 11
Thank you for pointing this out. We deleted one ‘the’ and have critically gone through the manuscript and have changed any textual mistakes.
Comment 12
Page 4, line 197: I suggest to insert after ‘sleep;’: the duration intervals of these.
Response 12
We adjusted the text to:
Materials and Methods, Data analyses: ”… and one part for the time spent in sleep; these five…”
Comment 13
Page 4, line 198: where is the 5-part composition illustrated?
Response 13
For the different thresholds of physical activity, the distributions of the 5-part compositions are illustrated in Figure 2a-c. We do not refer to Figure 2a-c in the Materials and Methods section; we do so in the Results under the heading ”Exposure compositions”.
The text in the Materials and Methods has been clarified as follows:
Materials and Methods, Data analyses: “The distributions of each of the parts of the 5-part compositions were illustrated by standard cumulative distributions.”
Comment 14
Page 5, lines 232 – 241: the description of the method of visualization by heat maps could be better placed in a separate section (in the methods chapter) about descriptive visualization. Particularly the use of the word ‘simulation’ should be avoided in a section about data-analysis.
Response 14
We placed our methods on visualisation of the regression equations by heat maps in a separate section. It is now under the sub-heading ”Visualisation of the regression analyses”, which is now the last sub-heading of the Materials and Methods section.
Comment 15
Page 8, in the capture of Figure 2 you say ‘probability/distributions’, while you display - I think – proportion scores. The distributions are assumptions to be tested. To me, this is confusing, as if you replace assumptions with data outputs.
Response 15
To clarify this, we have adjusted the figure caption to ”Cumulative distribution functions for the participants’ (n=72) percentage of time in sleep and in activity below and above (a) 20 %HRR, (b) 30 %HRR, (c) 40 %HRR during work and leisure.”
Comment 16
Page 11, figure 3 shows why I argue about replacing your multiplicative version of your metrics ilr1 and ilr2 by a different version: the heat maps invoke a rich view on deeper results from your data. I think I scratched only the surface with my suggestion for a (profile) Chi square test.
Response 16
We agree that the heat maps illustrate what appears to be quite different associations, even though the outcome – pain intensity – is the same in all three panels. We believe, referring to the explorative nature of the study, that this is a result of the three panels illustrating three different regression models, if with the same outcome.
Comment 17
Pages 11 – 19: incomplete references with missing page numbers are abundant. For instance: references , 64, 72, 73, 75.
Response 17
Thank you for pointing this out. We have gone through the references and have added page numbers and other information where it was missing.
Comment 18
Page 13, line 366, ‘solving’ should read: ‘fitting’.
Response 18
Equations are solved in this case, rather that fitted, since the text refers to illustrating the finalized regression equations; therefore, we chose to keep “solving” in this sentence.
Comment 19
Page 19, lines 640 – 642: Maybe reference 47 needs scrutiny? It seems incomplete.
Response 19
We have gone through the references and made adjustment where needed.
Comment 20
Page 19, line 646: inconsistent lay-out: year 2003 not written bold. Maybe reference 49 needs scrutiny?
Response 20
This reference, i.e. Astrand et al. (2003), is a book, and according to the MPDI reference style the year should not be written in bold. However, this reference indeed needed scrutiny, so thank you for pointing this out.
Comment 21
Page 19, lines 665: Maybe reference 57 needs scrutiny? It seems incomplete.
Response 21
Thank you for pointing this out. This is a chart/table published online by Health Profile Institute in Sweden. We have changed the reference type from ‘chart/table’ to a ‘webpage’ so that reader can easily access the chart/table.
Reviewer 3 Report
The manuscript consists in an exploratory study focused on to analyse the relation between physical activity at work and leisure time, and the musculoskeletal pain, including construction and healthcare workers. The authors used a %Heart rate metric and a questionnaire to quantify the musculoskeletal pain.
My main concern is related to the questionnaire applied to measure the musculoskeletal pain. The authors should explain better the structure of the questionnaire in the methodology. Did you apply the Nordic Questionnaire? This is a validated tool for this purpose. However, It seems that the authors did some changes in this tool, and it must be explained and justified.
Author Response
Thank you kindly for reviewing our manuscript.
Regarding your concerns related to the questionnaire applied to measure musculoskeletal pain, we did not apply the Nordic Questionnaire. However, the same body regions as those in the Nordic questionnaire were assessed and we used the drawing of the mannequin from the Nordic questionnaire. The structure of the questionnaire has now been better described in the Materials and Methods section as follows:
Materials and Methods, 2.2. Dependent variable: overall musculoskeletal pain: ”The pain intensity question from the Pain Severity Index assessed the intensity of “Pain during the previous four weeks” on a 4-point scale (0 ‘no pain’, 1 ‘mild pain’, 2 ‘moderate pain’, 3 ‘severe pain’) [47]. Such categorical scales with verbal descriptors have been recommended when assessing pain intensity [48]. The pain intensity question for the neck/shoulder and upper limbs is similarly related to clinical signs as the Pain Severity Index in those two regions [47]. In the present study, the nine body regions from the ”Standardised Nordic questionnaire on musculoskeletal symptoms” were assessed, i.e. neck, left shoulder, right shoulder, elbows/wrists/hands, upper back, lower back, hips, knees, and ankles/feet [49]. The localization of the body regions was facilitated by a mannequin drawing from the ”Nordic questionnaire” [49].
Round 2
Reviewer 2 Report
Dear authors, I appreciate your well-thought comments.
I receive your comments and revisions as well, but I still
hesitate about the full applicability of the CoDA method,
in your research. I do think the method is mathematically
flawed, because of the multiplicative character of the estimator.
You could do better, I think.
Author Response
We thank you kindly for reviewing the manuscript again, and for accepting our comments and revisions. We also appreciate your critical thoughts regarding the data processing method applied in CoDA, as it is essential in research to be critical. However, we stick to our CoDA approach, affirmed by an extensive body of literature, arguing for the appropriateness and mathematical correctness of its use on our type of data, i.e. data that add up to a constant (100%) and are, therefore, restricted and correlated. Specifically, methods for calculating ilrs are prescriptive and cannot be changed (e.g. [1-3]). We therefore apply CoDA according to these accepted principles, which include the multiplicative properties of the ilrs. We look forward to learning about other methods than CoDA that appropriately address the highly correlated nature of our type of data.
References
- Filzmoser, P.; Hron, K.; Templ, M. Applied Compositional Data Analysis - With Worked Examples in R; Springer, Cham: 2018; p. 280.
- Pawlowsky-Glahn, V.; Buccianti, A. Compositional Data Analysis - Theory and Applications; Wiley: Southern Gate, Chichester, West Sussex, United Kindom, 2011; p. 378.
- Pawlowsky-Glahn, V.; Egozcue, J.J.; Tolosana-Delgado, R. Modeling and analysis of Compositional Data; Wiley: Southern Gate, Chichester, West Sussex, United Kindom, 2015; p. 273.
Reviewer 3 Report
The authors improved the manuscript, answering to the reviewer's doubts.
Author Response
Thank you for reviewing the manuscript again. We appreciate the time and effort you spent on it.